# PolyX2: Fast Detection of Homorepeats in Large Protein Datasets

**DOI:** 10.3390/genes13050758

**Published:** 2022-04-25

**Authors:** Pablo Mier, Miguel A. Andrade-Navarro

**Affiliations:** Institute of Organismic and Molecular Evolution, Johannes Gutenberg University Mainz, 55128 Mainz, Germany; andrade@uni-mainz.de

**Keywords:** low-complexity regions, homorepeats, web tool

## Abstract

Homorepeat sequences, consecutive runs of identical amino acids, are prevalent in eukaryotic proteins. It has become necessary to annotate and evaluate this feature in entire proteomes. The definition of what constitutes a homorepeat is not fixed, and different research approaches may require different definitions; therefore, flexible approaches to analyze homorepeats in complete proteomes are needed. Here, we present polyX2, a fast, simple but tunable script to scan protein datasets for all possible homorepeats. The user can modify the length of the window to scan, the minimum number of identical residues that must be found in the window, and the types of homorepeats to be found.

## 1. Introduction

Homorepeats, polyX or amino acid repeats, are protein motifs defined as consecutive runs of a single residue [1]. They are the simplest low-complexity region and rely on a highly localized abundance of an amino acid. Any amino acid can form them, although some polyX are much more abundant than others (which also depends on the species taxonomy) [2]. They are associated with multiple functionalities, from aiding protein localization to mediating protein–protein interactions [3]. Many homorepeats have not been functionally characterized yet, despite their high abundance in eukaryotes (15% of proteins contain at least one of them) [4]. This makes them an interesting protein motif for a post-sequencing step in terms of proteome annotation.

There are a few attempts in the literature to describe sets of homorepeats in protein datasets [5,6,7,8]. However, these approaches differ in their definitions of homorepeats: the minimum length to be considered and whether they should be pure or allowed to include other amino acids. Furthermore, precomputed datasets are either not available for download [5,8] or limited to model organisms [7]. Lastly, there is no available code or web tool to look for homorepeats from scratch in a given protein dataset of interest.

To address these issues, we developed polyX2, a script (also available as a web tool) to compute the homorepeats of a given protein dataset, allowing the user to define length and purity thresholds, and producing results for all or a subset of homorepeat types.

## 2. Implementation

The script takes as input a set of protein sequences in FASTA format (Figure 1). It scans them with a window of a given length (default = 10 amino acids). A second parameter is the minimum number of identical residues required in the window to consider it to be part of a homorepeat (default = 8). Both default values have been used in previous research [2,9]. It results in parameter *k*, the maximum number of guest amino acids allowed in a window (default = 10 − 8 = 2). By definition, *k* must be smaller than half the window size. Per window, the following process is followed, starting at the first amino acid of the window:Tally amino acid occurrence. Move on to the next position.Exit the processing of the window if the amino acid count (ignoring the most frequently encountered amino acid) exceeds *k*. Otherwise, go back to step 1. This step greatly speeds up the procedure because it is often not necessary to examine a window all the way to find that it does not contain a homorepeat.

The sequence covered by the window is annotated as being part of a homorepeat formed by the most abundant amino acid if the number of guest amino acids *k* was not reached. The window is then shifted one position and the procedure is repeated. The homorepeat sequence may be extended by consecutive window positions until a window does not detect the homorepeat or the protein sequence is exhausted. The terminals of the resulting covered sequence are trimmed to remove all consecutive amino acids of the non-repetitive type so that the resulting sequence starts and ends with the most frequent amino acid.

The protein ID, start and end coordinates of the homorepeat, type of homorepeat, and sequence of the homorepeat are stored. To optimize both the execution time and memory usage, the results are initially saved in memory and then stored in the output file in batches of 10,000 homorepeats.

The script is written in the Perl language and has the following dependencies: Perl ≥ v5.28.0 and BioPerl library Bio:SeqIO.

## 3. Results

The script can be either executed in the web tool polyX2 (http://cbdm-01.zdv.uni-mainz.de~munoz/polyx2/; accessed on 28 March 2022) or downloaded to be run locally. In both cases, it only requires as input a set of proteins in FASTA format. The default parameters of a minimum number of 8 identical residues in a local window of 10 amino acids (≥8/10) can also be modified. The minimum must be larger than half the size of the window, and not larger than the window length; otherwise, default parameters are forced. As input examples in the web tool, the protein sequence for human huntingtin (UniProtKB:P42858) and the complete proteome of the SARS-CoV-2 virus are available to fill the input area (Figure 2a). By default, the script looks for all homorepeats; however, the user can speed up the search by limiting it to only some homorepeats of interest. The results are tabulated per homorepeat type, and a raw file with the homorepeat sequences located in the input protein dataset is produced as output (Figure 2b). The purity of the homorepeat is also denoted (number of residues of amino acid X versus the homorepeat length), as well as the guest residues found.

We analyzed several datasets both locally and in the web server, with default thresholds (Table 1; sets of homorepeats found per dataset are available on the website). The script processes, on average, around 1280 proteins per second (standalone). In the server, the process is three times slower due to dataset upload and server congestion; once a dataset is provided, the user receives a URL in which the results will be shown when the execution is completed.

Dataset upload is limited to 50 Mb (as a reference, the human proteome is 13.8 Mb), and therefore SwissProt v2020_06 (279.2 Mb) cannot be processed in the server. We advise using the standalone version of the script for datasets larger than 40,000 proteins.

The execution time is not linearly correlated with the number of proteins but with the number of amino acids in the input dataset (longer proteins require more window processing). Less stringent thresholds and shorter windows lead to higher *k* values and more windows, respectively, which is reflected in longer execution times.

We previously developed dAPE (http://cbdm-01.zdv.uni-mainz.de/~munoz/polyx; accessed on 28 March 2022) [8], a web tool to assess the evolution of homorepeats and their protein context. dAPE is more focused on the evolution of the polyX regions than on their search, and it is intended to be used with a set of (up to 20) orthologous proteins as input. Differently, PolyX2 can handle large protein datasets in a reasonable execution time, treating the proteins independently.

The PolyX2 tool allows the search for homorepeats in a simple yet fast way. We believe it is a useful resource given the customizable query (length and purity thresholds, and selection of homorepeat type), plus the duality of running either on our server or locally. We conclude that it would be easily integrated into an annotation pipeline after the sequencing of a new proteome as a step to characterize its homorepeats.

## Figures and Tables

**Figure 1 genes-13-00758-f001:**
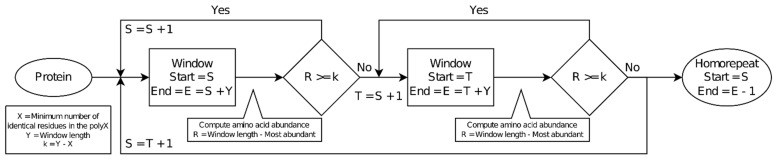
PolyX2 workflow.

**Figure 2 genes-13-00758-f002:**
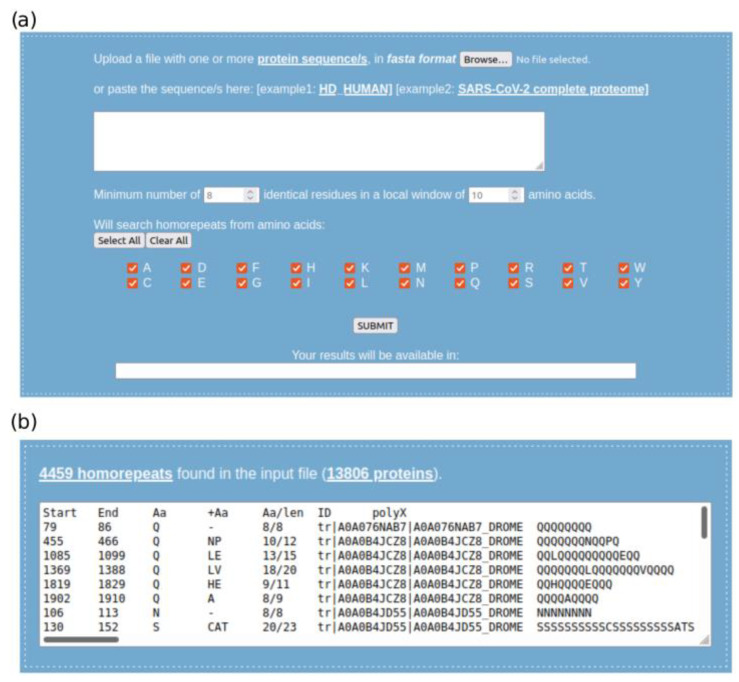
PolyX2 web tool. (**a**) Execution module on home page; (**b**) overview of the results.

**Table 1 genes-13-00758-t001:** Running time and number of homorepeats detected for several protein datasets and default parameters in the web server and using the standalone script.

Datasets (from UniProt v2020_06)	Number of Proteins	Time ^1^ (Web Server ^2^)	Time ^1^ (Standalone ^3^)	Number of Homorepeats
Fruit fly proteome	13,806	30 s	12 s	4459
Human proteome	20,609	46 s	21 s	2871
Isoform sequences	40,403	1 m 33 s	47 s	6303
SwissProt	563,972	-	6 m 59 s	26,535

^1^ Averaged after 10 executions. ^2^ Server running Debian GNU/Linux 10. ^3^ Executed on a Lenovo ThinkPad 64-bit with 15.3 Gb of RAM and an Intel Core i7-8665U CPU @ 1.90GHz × 8, running Ubuntu 20.04.02 LTS.

## Data Availability

Not applicable.

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
