# Peer review of "PolyX2: Fast Detection of Homorepeats in Large Protein Datasets"

_genes, 2022, doi:10.3390/genes13050758_

Round 1

Reviewer 1 Report

The paper of Mier and Andrade-Navarro describes a software script, polyX2, to detect homorepeats in amino acid sequences. The script works as expected and the manuscript is clearly written. I’ve a few concerns regarding the usability of the web interface and the interpretation of the parameters.

The web interface look & feel is really old-styled, missing the functionality of the Web 2.0. In the home page, the form is centered but section titles are left aligned, causing visualization problems in large screens. I would recommend using a CSS library to manage layout (i.e. to create a container that keeps the content centered) and responsiveness to window resizing. I recommend “Bootstrap CSS” (https://getbootstrap.com/) which is super easy to integrate and understand.

I would use the same input page for the results, without opening a new tab.

I have only one comment about the input parameters and how they are described in the manuscript. You defined a parameter k which is the maximum number of guest amino acids in the processed window and then said it has to be smaller than half of the window size. This is correct of course but in the interface you instead provide a check on the length of identical residues (which is complementary). I would change the manuscript to be coherent with the web interface. I think the user has in mind the repeat when using the tool.   

Author Response

Question 1 // The web interface look & feel is really old-styled, missing the functionality of the Web 2.0. In the home page, the form is centered but section titles are left aligned, causing visualization problems in large screens. I would recommend using a CSS library to manage layout (i.e. to create a container that keeps the content centered) and responsiveness to window resizing. I recommend “Bootstrap CSS” (https://getbootstrap.com/) which is super easy to integrate and understand.

Reply 1: We have modified the layout of the homepage (and output and error page) as per the reviewer’s suggestion.

Q2 // I would use the same input page for the results, without opening a new tab.
R2: We want to separate the inputs and the outputs, so after clicking the Submit button the results load on the same page, but the arrangements of the sections change; results are not shown in a different tab. The results include a section “Execute again” to be able to do a new search after the results of a search are shown. We have now included the “Help” section below the output as well, so as not to lose this information.

Q3 // I have only one comment about the input parameters and how they are described in the manuscript. You defined a parameter k which is the maximum number of guest amino acids in the processed window and then said it has to be smaller than half of the window size. This is correct of course but in the interface you instead provide a check on the length of identical residues (which is complementary). I would change the manuscript to be coherent with the web interface. I think the user has in mind the repeat when using the tool.
R3: The reviewer is right, we had used k (maximum number of guest amino acids) in the manuscript and the minimum number of identical residues in the web tool. As the reviewer states, they are complementary, but this could eventually be misleading. We have decided to keep the manuscript as is, and include an explanation about k in the server. The reason for this is that our pipeline uses indeed the k parameter, and we believe it is clearer to explain the method with it. Hence, we have modified the “Help” in the server to include k, and now the error page when the selected thresholds are not correct also includes an explanation about it.

Reviewer 2 Report

Comments for ” PolyX2: fast detection of homorepeats in large protein datasets”

  1. The biological meaning of homorepeats should be emphasized in the introduction.
  2. Authors developed a web tool for homorepeats in a previous study called dAPE, so it could be better if authors descript the differences between PolyX2 with dAPE and advantages of PolyX2.
  3. The case study is important to help readers understand the calculation results, if the case study is not appropriate in the manuscript due to page number limited, the case study could be available on the web page.
  4. In the Implementation section, authors describe their method to detect homorepeats, it would be better if authors can present the method as an algorithm form also.

Author Response

Question 1 // The biological meaning of homorepeats should be emphasized in the introduction.
Reply 1: We have expanded the initial paragraph in the Introduction discussing about homorepeats.

Q2: Authors developed a web tool for homorepeats in a previous study called dAPE, so it could be better if authors descript the differences between PolyX2 with dAPE and advantages of PolyX2.
R2: We have included a complete paragraph on this matter in section “Results”.

Q3: The case study is important to help readers understand the calculation results, if the case study is not appropriate in the manuscript due to page number limited, the case study could be available on the web page.
R3: As case studies, we analyzed several datasets both in the web server and locally using the available standalone code. We report the number of input proteins, number of found homorepeats and the execution times in Table 2. We comment them in the manuscript. All these input and output files are in the web server, in section “Precomputed”. 
Furthermore, in the web server we have two datasets to be used as input examples “HD_HUMAN” and “SARS-CoV-2 complete proteome”, as well as a part within the “Help” section commenting the output results for the protein “HD_HUMAN” and all columns in the output file. All of this is explicitly written in the manuscript.

Q4: In the Implementation section, authors describe their method to detect homorepeats, it would be better if authors can present the method as an algorithm form also. 
R4: Following this suggestion, we now present the method also depicted in a workflow (Figure 1).

Round 2

Reviewer 1 Report

The authors have improved both the manuscript and the web interface.